# ClusMol: Cluster-Assisted Contrastive Learning for Molecular Representations

## Abstract

Effective molecular representations are crucial for tasks like molecular property prediction and de novo molecular design. While deep learning has shifted molecular representation learning from hand-crafted fingerprints to learned features, the high cost of labeled data has motivated self-supervised learning (SSL) on large-scale unlabeled datasets, with contrastive learning being a leading approach. However, existing methods rely on augmented views of the same molecule as positives, overlooking structural and functional similarities between distinct molecules. To address this, we propose the Functional Group-Guided Molecular Graphs (FGMG) algorithm, which constructs molecular graphs with linear complexity while better capturing structure-activity relationships (SAR). Based on FGMG, we introduce ClusMol, a cluster-assisted contrastive learning pretraining framework that assigns graph-labels via graph clustering and employs a Chemical Semantic-Weighted Contrastive Loss (SemCoL) to treat molecules within the same cluster as "soft positive" pairs. Pretrained on three million molecules, ClusMol significantly outperforms state-of-the-art SSL baselines across downstream tasks, demonstrating the benefit of combining functional group–aware graph construction with cluster-assisted contrastive learning.

## 1 Introduction

Recent advances in artificial intelligence and computational methods have profoundly reshaped molecular science, opening new avenues for drug discovery, sustainable material design, and beyond (Deng et al. (2023); Mak et al. (2024)). A core challenge is the transformation of molecular structures into continuous, information-rich vectors, a process known as Molecular Representation Learning. The quality of this process fundamentally determines the performance limits of downstream tasks (Wigh et al. (2022)).

Early efforts in molecular representation relied heavily on hand-crafted features derived from domain knowledge. Examples include MACCS keys (Durant et al. (2002)), which encode the presence of predefined substructures as fixed binary vectors, and Morgan fingerprints (Morgan (1965)), whose iterative neighborhood-based algorithm later gave rise to the widely used Extended Connectivity Fingerprints (ECFP) (Rogers & Hahn (2010)). These descriptor achieved considerable success in QSAR (Quantitative Structure–Activity Relationship) tasks (Zhang et al. (2017); Ma et al. (2015)), but they were inherently fixed, low dimensional, and potentially biased. More importantly, such traditional features did not capture the full molecular topological and three-dimensional information, limiting their generality and expressiveness.

With the advent of deep learning, molecular representation learning has shifted toward data-driven approaches. Sequence based models, such as RNN (Elman (1990)), LSTM(Hochreiter & Schmidhuber (1997)) and transformer (Vaswani et al. (2017)), operate on string representations like SMILES (Weininger (1988)), SELFIES (Krenn et al. (2020)), while graph-based models, including GCN (Kipf (2016)), GIN (Xu et al. (2018)), and MPNN (Gilmer et al. (2017)) directly learn from molecular graphs. Despite their success in supervised molecular property prediction (Maron et al. (2019); Mansimov et al. (2019)), these models rely heavily on costly labeled data. In contrast, massive unlabeled databases like PubChem (Kim et al. (2025)) and ZINC (Irwin et al. (2012)) motivate

---

[0]Parts of this manuscript were drafted with the assistance of AI-based writing tools.

the use of self-supervised learning (SSL) to pre-train expressive molecular representations before fine-tuning on smaller labeled tasks (Zeng et al. (2022); Liu et al. (2021b)).

In self-supervised learning, contrastive methods have attracted significant attention due to their performance (Guan & Zhang (2023); Pinheiro et al. (2022)). They rely on instance discrimination: creating positive sample pairs via data augmentation while treating other molecules in the batch as negatives. The learning objective pulls positive pairs closer and pushes negative pairs apart in the representation space, resulting in robust embeddings insensitive to minor structural changes.

Despite its success, contrastive learning faces a fundamental challenge in chemistry: the "false negative problem" (Chuang et al. (2020)). As seen in Figure 1(a), standard contrastive learning pushes apart any two different molecules regardless of their underlying similarity (e.g., n-butane and isobutane), ignoring intrinsic chemical relationships. From a chemical perspective, homologues or compounds sharing pharmacophores should occupy similar positions in representation space rather than being treated as negatives. However, conventional contrastive learning forces the model to expend capacity on distinguishing molecules that ought to be recognized as similar, hindering its ability to learn chemically meaningful representations.

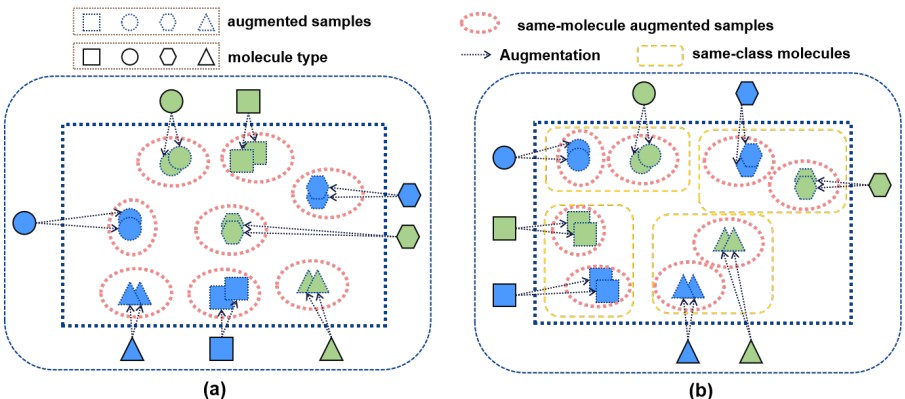

Figure 1: The contrast between traditional contrastive learning and CLUSMOL. (a) Traditional contrastive learning considers only augmented samples from the same molecule as positives, which forces structurally or functionally related molecules apart in the representation space. (b) CLUSMOL brings chemically similar molecules closer in the representation space via supervised signals derived from the Functional Group-Guided Molecular Graph (FGMG).

To address this false negative problem in contrastive learning, we propose the Cluster-Assisted Contrastive Learning (CLUSMOL) framework, which augments standard contrastive learning with supervisory signals extracted from a molecular relationship graph. Our key insight is that the vast chemical space exhibits an intrinsic "community structure", where molecules cluster based on shared structural, physical, or chemical properties (Figure 1(b)). CLUSMOL leverages this structure to guide representation learning. Specifically, we first introduce the Functional Group-Guided Molecular Graph (FGMG), a *linear-complexity* method for constructing large-scale molecular graphs that capture structure-activity relationships (SAR). We then apply graph clustering to partition this graph into chemical communities and assign community-based labels. Finally, we design the Chemical Semantic-Weighted Contrastive Loss (SemCoL) that treats molecules within the same community as "soft positives". In essence, CLUSMOL operates in three stages: (1) constructing a molecular relationship graph via FGMG, (2) clustering the graph to derive community labels, and (3) training with SemCoL. This reformulates the learning objective from simple "instance discrimination" into a chemically meaningful goal of intra-community aggregation and inter-community separation.

Our main contributions include the following.

1) We identify and articulate the "false negative problem" as a critical limitation of conventional contrastive learning in the chemical domain.

2) We propose the FGMG algorithm, a computationally efficient method for capturing chemically relevant molecular relationships at scale.

3) We introduce the CLUSMOL framework, a novel SSL paradigm that leverages graph clustering to generate supervisory signals based on molecular community structures.

4) We design the Chemical Semantic-Weighted Contrastive Loss (SemCoL), a flexible loss function that effectively integrates these community-based graph-labels into the contrastive learning process.

5) We demonstrate that CLUSMOL achieves state-of-the-art performance, significantly outperforming existing SSL baselines on multiple MoleculeNet benchmark tasks.

## 2 PRELIMINARIES

This section introduces key concepts, including GNNs and contrastive learning.

### 2.1 GRAPH NEURAL NETWORKS (GNNS)

GNNs are a class of deep learning models that learn node representations through an iterative message passing mechanism at layer $k$:

$$\mathbf{a}_v^{(k)} = \text{AGGREGATE}^{(k)}(\{\mathbf{h}_u^{(k-1)} : u \in \mathcal{N}(v)\}) \tag{1}$$

$$\mathbf{h}_v^{(k)} = \text{COMBINE}^{(k)}(\mathbf{h}_v^{(k-1)}, \mathbf{a}_v^{(k)}) \tag{2}$$

Here $\mathbf{h}_v^{(k)}$ is node $v$'s representation at layer $k$, $\mathcal{N}(v)$ are neighbors of $v$, AGGREGATE pools neighbor information, and COMBINE updates representations. After $K$ layers, a READOUT function produces the graph representation:

$$\mathbf{h}_G = \text{READOUT}(\{\mathbf{h}_v^{(K)} : v \in \mathcal{V}\}) \tag{3}$$

### 2.2 CONTRASTIVE LEARNING

Contrastive learning is a powerful self-supervised learning paradigm that aims to learn meaningful feature representations by distinguishing between "similar" samples (positive pairs) and "dissimilar" samples (negative pairs). Given an anchor sample $x_i$, a positive sample $x_j$ is constructed. All other samples in the batch, $\{x_k\}_{k \neq i}$, are treated as negative samples. The core objective of contrastive learning is to maximize the agreement between the anchor and its positive sample in the representation space while minimizing its agreement with all negative samples. This objective is typically achieved via a contrastive loss function, the most common of which is the InfoNCE (Noise Contrastive Estimation) loss, or its variant NT-Xent used in frameworks like SimCLR (Chen et al. (2020)). For a given positive pair $(i, j)$, the loss function is formulated as:

$$\mathcal{L}_{i,j} = -\log \frac{\exp(\text{sim}(\mathbf{z}_i, \mathbf{z}_j)/\tau)}{\sum_{k=1, k \neq i}^{2N} \exp(\text{sim}(\mathbf{z}_i, \mathbf{z}_k)/\tau)} \tag{4}$$

where $\mathbf{z}_i$ and $\mathbf{z}_j$ are the final representation vectors of the positive pair, obtained after passing through an encoder and a projection head. A batch of $N$ samples yields $2N$ augmented views. $\text{sim}(\mathbf{u}, \mathbf{v}) = \frac{\mathbf{u}^\top \mathbf{v}}{\|\mathbf{u}\|\|\mathbf{v}\|}$ is the cosine similarity between two vectors. $\tau$ is a temperature hyperparameter that scales the distribution of similarities. A lower temperature helps the model learn from hard-to-distinguish negative samples.

## 3 CLUSMOL

This section introduces the overview of CLUSMOL framework and its core components.

### 3.1 FRAMEWORK OVERVIEW

The overall CLUSMOL framework, illustrated in Figure 2, consists of three main stages.

**Graph Construction**: Starting from an unlabeled molecular dataset, the proposed FGMG algorithm models the relationships between molecules, constructing a global molecular relationship graph.

**Graph-Label Generation**: An efficient community detection algorithm is applied to partition the graph into clusters, assigning each molecule a discrete graph label that corresponds to its identified chemical community.

**Pre-training**: The generated graph labels serve as supervisory signals for contrastive pre-training. Each molecule is augmented twice to create two distinct views, which are then fed into a GNN encoder. The encoder is trained by minimizing the proposed SemCoL function, which encourages the model to group augmented views of the same source molecule while also pulling together representations of molecules that share the same graph label in the embedding space.

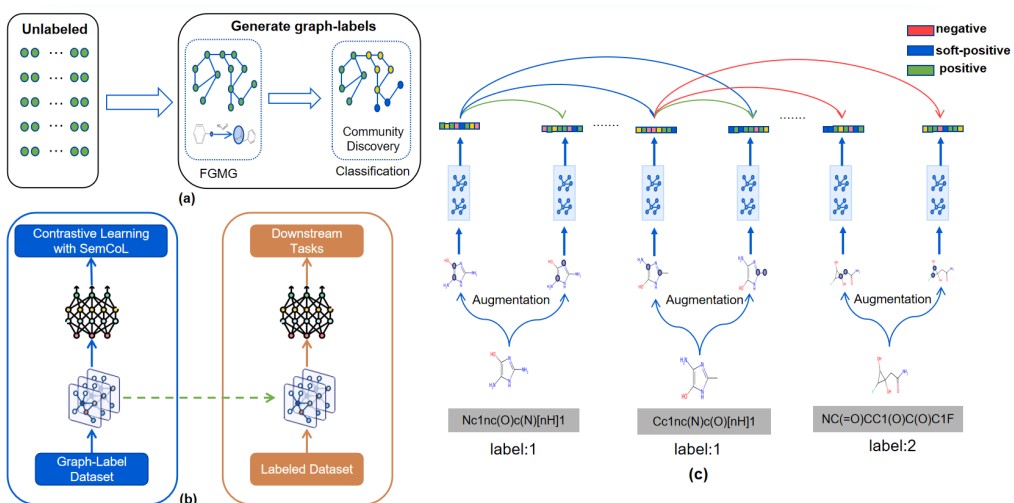

Figure 2: Overview of CLUSMOL. (a) Generate graph-labels for the unlabeled dataset by FGMG. (b) The whole training process of CLUSMOL. GNNs are first pre-trained to learn representative features. For downstream tasks, the pre-trained GNN encoder is shared and an MLP head is randomly initialized, followed by supervised fine tuning. (c) SemCoL introduces soft positive samples into contrastive learning to better capture chemical relationships.

### 3.2 MOLECULAR RELATIONSHIP GRAPH CONSTRUCTION

To apply clustering algorithms to molecular datasets, it is first necessary to construct a global graph that captures intrinsic relationships between molecules. An intuitive approach is to define edges based on structural similarity, such as the Tanimoto coefficient (Willett et al. (1998)). However, this purely similarity-based approach face two key limitations:

- **Ambiguity in Chemical Meaning:** It may erroneously cluster "Activity Cliffs" (Maggiora (2006)) molecules that are structurally similar but exhibit vastly different biological activities, or separate molecules with different chemical scaffolds but similar functions, thereby introducing misleading supervisory signals.
- **Computational Complexity:** For a dataset with $M$ molecules, calculating all pairwise similarities requires $O(M^2)$ time, which is computationally prohibitive at large scale.

To overcome the above limitations, we introduce the Functional Group-Guided Molecular Graph (FGMG), which formalizes molecular relationships based on explicit chemical transformations rather than abstract similarity metrics.

**Definition 1** (Functional Group-Guided Molecular Graph, FGMG). *An FGMG is defined as a molecular relationship graph $G = (\mathcal{M}, \mathcal{E})$, where the node set $\mathcal{M}$ represents all molecules in the dataset $\mathcal{D} = \{m_1, m_2, \ldots, m_M\}$, and the edge set $\mathcal{E}$ captures feasible chemical modification relationship between molecules.*

*Specifically, let $\mathcal{F} = \{f_1, f_2, \ldots, f_K\}$ denote a set of predefined functional groups. For any two molecules $m_i, m_j \in \mathcal{D}$, an undirected edge $(m_i, m_j) \in \mathcal{E}$ exists if and only if one molecule can be obtained from the other through a single-step hydrogen atom substitution reaction:*

$$\exists f_k \in \mathcal{F}, \quad s.\ t. \quad m_j \cong Substitute(m_i, H, f_k)$$

*Here, $Substitute(m_i, H, f_k)$ denotes substituting a hydrogen atom (H) in molecule $m_i$ with the functional group $f_k$, and $\cong$ denotes molecular structure isomorphism.*

The detailed construction algorithm of FGMG is provided in Appendix A.

**Interpretation and Advantages.** FGMG provides several significant advantages.

1) *Chemical Relevance:* Each edge represents a valid chemical transformation, and clusters identified via community detection correspond to sets of molecules that can be interconverted through a series of functional group substitution. These clusters tend to share a common "core structure".

2) *Structure-Activity Relationship (SAR):* This clustering process naturally captures principles of chemical synthesis and molecule accessibility, highlighting how minor changes in functional groups systematically affect molecular properties. This yields chemically meaningful graph labels for downstream molecular property prediction models.

3) *Computational Efficiency:* Unlike similarity-based approaches requiring $O(M^2)$ pairwise comparisons, the FGMG construction algorithm scales with the number of molecules $M$, the average number of hydrogen atoms per molecule $H$, and the size of the functional group set $K$, resulting in a time complexity of $O(MHK)$. In practice, this achieves near-linear scalability with respect to the dataset size $M$, enabling efficient construction even for huge molecular datasets.

**Graph Storage and Reusability.** To further improve scalability and usability, the constructed FGMG can be stored in graph databases such as Neo4j. This brings two important benefits:

1) *Incremental Construction.* New molecules can be added into the existing graph without rebuilding the entire structure, supporting continuous expansion as new chemical data becomes available.

2) *Reusability Across Applications*. A single constructed FGMG can be accessed multiple times by different downstream tasks (e.g., molecular property prediction, representation learning), avoiding redundant preprocessing and ensuring consistent graph-label annotations.

In short, FGMG combines chemical interpretability, computational scalability, and practical reusability, making it a versatile foundation for large-scale molecular representation learning.

### 3.3 GRAPH-LABEL GENERATION

The constructed molecular relationship graph FGMG captures local connectivity within chemical space, yet higher-level community structures remain implicit. Our objective is to discover these "chemical communities" of structurally or functionally similar molecules using clustering algorithms, and to leverage them as prior knowledge for subsequent contrastive learning.

**Clustering Algorithm.** On the constructed FGMG graph G, we employ the Louvain community detection algorithm (Blondel et al. (2008)), an efficient, modularity-optimized hierarchical clustering algorithm. Modularity (Newman & Girvan (2004)) evaluates the quality of a network's community structure by quantifying the difference between the density of intra-community connections and the sparsity of inter-community connections. The Louvain algorithm was chosen for its outstanding computational efficiency and scalability, capable of processing graphs with millions of nodes, and has demonstrated strong performance across diverse domains.

**Label Generation.** Applying the Louvain algorithm, the molecular set $\mathcal{M}$ is partitioned into $K$ mutually exclusive clusters $\{C_1, C_2, \ldots, C_K\}$. Each molecule $m_i$ is then assigned a unique cluster ID $c_i \in \{1, 2, \ldots, K\}$, which serves as a pseudo-label for subsequent contrastive learning.

### 3.4 CONTRASTIVE LEARNING WITH GRAPH-LABELS

Through the cluster result in step two, we convert the unlabeled dataset into a dataset with graph-labels. We follow traditional contrast learning way, but use Chemical Semantic-Weighted Contrastive Loss (SemCoL) to substitute the traditional contrastive loss function like InfoNCE (Oord et al. (2018); Chen et al. (2020)).

**Data Augmentation.** Each molecule $m_i \in \mathcal{M}$ is independently and randomly augmented to obtain two different augmented views $v_i$ and $v_i'$, following procedures similar to MolCLR (Wang et al. (2022)). In CLUSMOL, we employ atom masking and bond deletion as augmentation strategies.

**Definition 2** (Positive Sample Pairs). *Two types of positive pairs are defined based on graph labels:*

*1) **Hard Positive Pairs**: Augmented view pair $(v_i, v_i')$ from the same molecule $m_i$, representing semantic equivalence.*

*2) **Soft Positive Pairs**: Augmented view pair $(v_i, v_j)$ from different molecules $(m_i, m_j)$ with same graph-label $(c_i = c_j)$, representing semantic similarity without equivalence.*

**Contrastive Loss Function.** To distinguish between these two positive samples during training, we designed a weighted contrastive loss function as follows.

**Definition 3** (Chemical Semantic-Weighted Contrastive Loss, SemCoL). *For a batch containing $N$ molecules, which yields $2N$ views after data augmentation, let $\{v_k\}_{k=1}^{2N}$ denote the features of all views. For any view $v_i$, the loss function $\mathcal{L}_i$ is defined as:*

$$\mathcal{L}_i = -\log \frac{\sum_{j \in \mathcal{P}(i)} w_{ij} \exp(sim(v_i, v_j)/\tau)}{\sum_{k=1, k \neq i}^{2N} \exp(sim(v_i, v_k)/\tau)}$$

*where:*

- $\mathcal{P}(i)$ *is the set of view indices in the batch that form positive sample pairs with view $i$.*

- $\tau$ *is a temperature hyperparameter.*

- $sim(\cdot, \cdot)$ *represents cosine similarity.*

- $w_{ij}$ *is the weight of positive sample pair $(i, j)$, defined as follows:*

$$w_{ij} = \begin{cases} 1 & \text{if } m_i = m_j \ \text{(hard positive sample)} \\ \alpha & \text{if } m_i \neq m_j \text{ and } c_i = c_j \ \text{(soft positive sample)} \end{cases}$$

*where $\alpha \in [0, 1)$ controls the influence of soft positives. Setting $\alpha = 0$ reduces SemCoL to the standard contrastive loss, while $\alpha > 0$ encourages the model to pull hard positives closely together and push soft positives from the same chemical community with lesser force. The overall pre-training loss $\mathcal{L}$ is the average of all $2N$ view losses in the batch.*

## 4 EXPERIMENTS

This section evaluates CLUSMOL in terms of datasets, baselines, and prototype implementation, followed by downstream performance results. We aim to answer:

> **RQ**: *Does* CLUSMOL, *leveraging FGMG and SemCoL, improve molecular representations and downstream task performance?*

Experiments are organized into three aspects: fine-tuning performance, linear probing performance, and ablation studies on key components.

### 4.1 EXPERIMENTAL SETTINGS

**Datasets.** Pre-training is conducted on the GDB-10 dataset (Blum & Reymond (2009)) with downstream evaluation following the MolecularNet benchmark (Wu et al. (2018)). Downstream tasks include five classification (BBBP, BACE, ClinTox, Tox21, Sider) and five regression (ESOL, FreeSolv, Lipo, QM8, QM9) tasks. For multi-task datasets like Sider and QM8, results are averaged across all subtasks. Detailed statistics for all downstream datasets are provided in Appendix B.

**Baselines.** CLUSMOL is evaluated against both supervised and self-supervised baselines. Supervised baselines include AttentiveFP (Xiong et al. (2019)), and D-MPNN (Yang et al. (2019)). Self-supervised baselines cover generative approaches like InfoGraph (Sun et al. (2019)), AttrMask,

ContextPred (Rong et al. (2020)), G-Motif (Chithrananda et al. (2020)) and KGG (To et al. (2025)), as well as contrastive methods like GraphCL (You et al. (2020)), GraphMVP (Liu et al. (2021a)), and MolCLR (Wang et al. (2022)). Detailed descriptions of all baselines are summarized in Appendix D.

**CLUSMOL Prototype.** Graph-labels for pre-training are generated by first constructing molecular relationship graphs using the proposed FGMG approach, followed by Louvain clustering to partition molecules into chemically meaningful communities. The resulting FGMG graphs are stored in a Neo4j graph database, enabling incremental construction and efficient reuse. Detailed statistics of the graph-label dataset extracted from GDB-10 are provided in Appendix C.

Dataset splits follow MoleculeNet standards (80% training, 10% validation, 10% test), and baseline methods adopt the hyperparameters recommended in their original works. The GNN backbone is a 5-layer GIN with a hidden dimension of 300 and an output dimension of 512. All experiments are repeated with three random seeds, and the mean and standard deviation are reported. Detailed training settings for CLUSMOL are provided in Appendix D.

## 4.2 FINE-TUNING PERFORMANCE

Table 1: CLUSMOL Performance on Molecular Property Prediction Tasks. The mean and standard deviation are reported. For each benchmark, the overall best performing method is highlighted in **bold**, and the best self-supervised/pre-trained method is marked in *bold.

| Metric | Classification (ROC-AUC %, higher is better ↑) | | | | | Regression (lower is better ↓) | | | | |
|---|---|---|---|---|---|---|---|---|---|---|
| Dataset | BBBP | BACE | ClinTox | Tox21 | Sider | ESOL | FreeSolv | Lipo | QM8 | QM9 |
| Molecules | (2039) | (1513) | (1478) | (7831) | (1427) | (1128) | (642) | (4200) | (21786) | (133885) |
| Tasks | (1) | (1) | (2) | (12) | (27) | (1) | (1) | (1) | (12) | (12) |
| D-MPNN | 0.707 | 0.763 | **0.902** | 0.754 | 0.583 | 0.630 | **0.780** | 0.554 | 0.032 | 2.942 |
| | (0.015) | (0.009) | (0.005) | (0.021) | (0.007) | (0.029) | (0.039) | (0.014) | (0.000022) | (0.017) |
| Attentive FP | 0.674 | 0.811 | 0.889 | **0.839** | 0.631 | **0.579** | 0.935 | 0.607 | 0.010 | 2.760 |
| | (0.006) | (0.037) | (0.037) | (0.013) | (0.021) | (0.036) | (0.224) | (0.026) | (0.000213) | (0.083) |
| AttrMask | 0.705 | 0.801 | 0.775 | 0.810 | 0.616 | 0.876 | 1.430 | 0.703 | 0.031 | 5.010 |
| | (0.035) | (0.032) | (0.083) | (0.023) | (0.016) | (0.102) | (0.314) | (0.024) | (0.001) | (0.230) |
| ContextPred | 0.674 | 0.801 | 0.767 | 0.808 | 0.625 | 0.892 | 1.460 | 0.702 | 0.031 | 5.240 |
| | (0.026) | (0.004) | (0.046) | (0.015) | (0.022) | (0.074) | (0.298) | (0.024) | (0.001) | (0.121) |
| InfoGraph | 0.613 | 0.675 | 0.720 | 0.773 | 0.620 | 1.010 | 2.000 | 0.806 | 0.033 | 5.780 |
| | (0.021) | (0.005) | (0.061) | (0.025) | (0.006) | (0.110) | (0.112) | (0.046) | (0.001) | (0.098) |
| G-Motif | 0.683 | 0.808 | 0.716 | 0.801 | 0.642 | 0.827 | 1.450 | 0.694 | 0.030 | 4.690 |
| | (0.049) | (0.015) | (0.055) | (0.021) | (0.036) | (0.084) | (0.162) | (0.039) | (0.001) | (0.143) |
| MolCLR | 0.699 | 0.813 | 0.857 | 0.811 | 0.614 | 0.884 | 1.520 | 0.682 | 0.012 | 8.510 |
| | (0.016) | (0.010) | (0.028) | (0.010) | (0.012) | (0.115) | (0.305) | (0.038) | (0.0001) | (0.678) |
| GraphCL | 0.707 | 0.754 | 0.744 | 0.754 | 0.645 | 0.854 | 1.410 | 0.714 | 0.030 | 6.850 |
| | (0.012) | (0.042) | (0.101) | (0.028) | (0.018) | (0.094) | (0.189) | (0.045) | (0.001) | (2.770) |
| GraphMVP | 0.671 | 0.766 | 0.750 | 0.761 | 0.631 | 0.830 | 1.410 | 0.704 | 0.029 | 5.110 |
| | (0.030) | (0.014) | (0.083) | (0.031) | (0.011) | (0.122) | (0.289) | (0.032) | (0.001) | (0.721) |
| KGG | 0.691 | 0.817 | 0.853 | *0.825 | 0.636 | 0.677 | 2.100 | 0.598 | 0.012 | 3.244 |
| | (0.022) | (0.017) | (0.018) | (0.004) | (0.008) | (0.027) | (0.138) | (0.005) | (0.0002) | (0.131) |
| | ROC-AUC | | | | | RMSE | | | MAE | |
| CLUSMol | **0.712** | **0.819** | *0.882 | 0.812 | **0.645** | *0.667 | *1.026 | 0.548 | **0.010** | **2.190** |
| | (0.004) | (0.016) | (0.004) | (0.006) | (0.012) | (0.015) | (0.061) | (0.035) | (0.0001) | (0.013) |

For all self-supervised methods, pre-trained models are fine-tuned on MoleculeNet tasks for 100 epochs. The checkpoint achieving the best validation performance is selected for test evaluation. For CLUSMOL, the weight $w$ in the SemCoL is set to 0.5.

**Results and Analysis.** We evaluate models using ROC-AUC (Receiver Operating Characteristic – Area Under the Curve) for classification tasks, and RMSE (Root Mean Square Error) and MAE (Mean Absolute Error) for regression tasks. As shown in Table 1, CLUSMOL outperforms all self-supervised baselines on nine tasks except Tox21, and surpasses supervised methods like D-MPNN and AttentiveFP on most tasks. These gain stem from incorporating intermolecular relationships during pre-training, which project chemically similar molecules closer in vector space. By combining supervised data utilization with self-supervised accessibility, supported by FGMG's efficient relationship extraction, CLUSMOL achieves strong generalization. Notably, it excels on small datasets (e.g., ClinTox, ESOL, Sider) and 3D-sensitive ones (QM8, QM9), capturing geometric insights without explicit 3D data. Detailed QM8 and QM9 results are in Appendix D.

## 4.3 LINEAR PROBING PERFORMANCE

To demonstrate that CLUSMOL learns chemically meaningful representations, the pre-trained encoder is frozen and its outputs are fed into a linear head for evaluation on seven datasets. Comparisons are made against MolCLR, AttrMask, GraphMVP, and KGG representations, as well as the same-dimensional Morgan fingerprints. All experiments are run over three random seeds, reporting means and standard deviations. Detailed per-task results for QM8 are provided in Appendix D.

**Results.** As shown in Table 2, CLUSMOL representations outperform other pre-trained methods on all nine tasks and exceed Morgan fingerprints on eight (except Lipo). Its strong results on QM8 confirm prior analysis. The minimal difference between fine-tuned and probing results indicates that CLUSMOL learns predictive representations during pre-training, consistent with FGMG's design.

Table 2: Linear Protocol Performance of CLUSMOL

|  | Classification (ROC-AUC %, higher is better ↑) | | | | | Regression (lower is better ↓) | | | |
|  | BBBP | BACE | ClinTox | Tox21 | Sider | ESOL | FreeSolv | Lipo | QM8 |
|---|---|---|---|---|---|---|---|---|---|
| AttrMask | 0.540(0.034) | 0.628(0.016) | 0.533(0.033) | 0.676(0.043) | 0.534(0.025) | 2.040(0.090) | 3.640(0.161) | 1.170(0.043) | 0.040(0.0002) |
| MolCLR | 0.683(0.019) | 0.517(0.008) | 0.752(0.020) | 0.752(0.019) | 0.593(0.007) | 1.550(0.001) | 2.420(0.136) | 1.140(0.004) | 0.023(0.002) |
| GraphMVP | 0.588(0.0003) | 0.609(0.002) | 0.510(0.014) | 0.698(0.028) | 0.528(0.028) | 1.670(0.118) | 3.140(0.204) | 1.090(0.043) | 0.040(0.002) |
| KGG | 0.586(0.015) | 0.734(0.016) | 0.764(0.009) | 0.712(0.003) | 0.570(0.003) | 1.346(0.017) | 3.481(0.071) | 1.199(0.005) | 0.027(0.001) |
| Morgan | 0.674(0.004) | 0.774(0.011) | 0.711(0.033) | 0.761(0.020) | 0.637(0.027) | 1.215(0.053) | 1.987(0.101) | **0.793(0.041)** | 0.017(0.001) |
| CLUSMOL | **0.692(0.008)** | **0.792(0.005)** | **0.807(0.009)** | **0.823(0.006)** | **0.642(0.023)** | **0.866(0.069)** | **1.497(0.071)** | 0.905(0.011) | **0.014(0.001)** |

## 4.4 ABLATION STUDIES

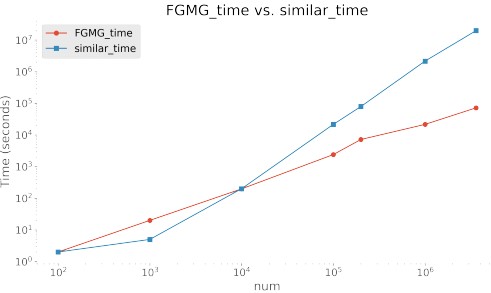

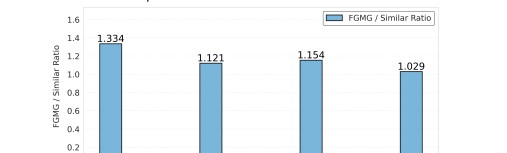

(a) FGMG vs. Similarity-based methods: Time consumption.

(b) Downstream performance comparison of FGMG and similarity-based graphs.

Figure 3: Comparison of FGMG and similarity-based approaches.

### 4.4.1 THE EFFECTIVENESS OF FGMG

We compare pre-training effects using graph-labeled datasets generated via similarity-based graphs versus FGMG. Figure 3a illustrates time consumption across dataset scales. Due to prohibitive costs for similarity graphs at the million-scale, we subsample 200,000 molecules from GDB-10 (similarity threshold 0.8). Models are pre-trained on respective datasets and evaluated downstream. Results in Figure 3b show FGMG yields superior performance, highlighting its efficiency and effectiveness in capturing molecular relationships.

### 4.4.2 THE INFLUENCE OF THE WEIGHT IN SEMCOL

To assess the impact of soft positive pair weights, models are pre-trained with $w = 0.0, 0.3, 0.5, 0.7$ and downstream performance is compared. As shown in Figure 4, most tasks exhibit an initial rise followed by a decline. When $w$ is too low, when equal to 0, it is equivalent to InfoNCE, it reverts to standard contrastive learning, limiting the incorporation of chemical knowledge. Excessive $w$ confuses soft positive and hard positive samples, contradicting the intended design of CLUSMOL.

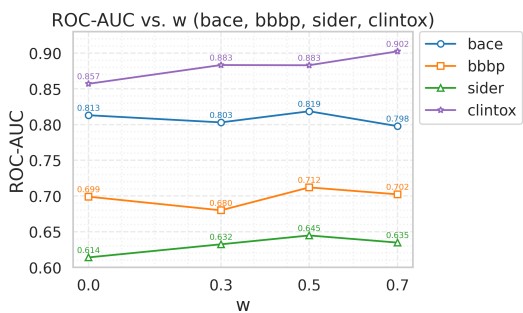

Figure 4: The impact of soft label weights on CLUSMOL performance.

## 4.5 VISUALIZATION OF MOLECULAR REPRESENTATIONS LEARNED BY CLUSMOL VIA T-SNE

To investigate the efficacy of pseudo-labels in pre-training and to verify CLUSMol's ability to capture molecular community information, we visualized the representation space learned by the encoder using t-SNE. Figure 5 illustrates the embedding space of 100,000 molecules randomly sampled from the pre-training dataset, where the data points are colored according to the pseudo-labels generated by clustering.

To further validate the rationality of the clustering results and the representational capacity of the encoder, we selected ten representative molecules from distinct clusters and annotated their specific coordinates within the vector space (as shown in Figure 5). Additionally, to assess CLUSMol's capability in pre-learning molecular properties, we evaluated its performance on the QM9 dataset, which features molecules with larger molecular weights. The resulting visualization, colored by specific properties, is presented in Figure 6a and Figure 6b.

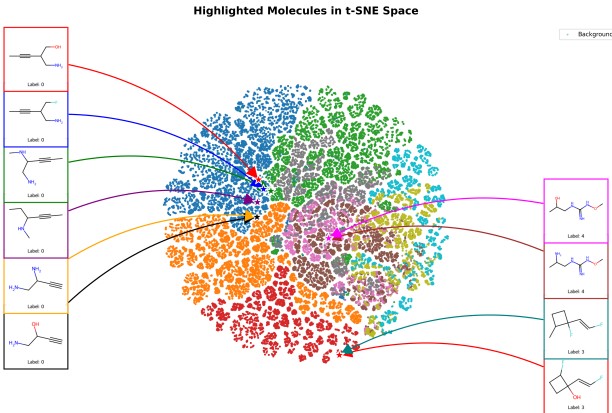

Figure 5: Visualization of molecular representations learned by CLUSMOL by labeled with pseudo-labels

## 5 RELATED WORK

**Molecular Representation Learning.** Molecular representation learning has evolved from hand-crafted molecular fingerprints (Fu et al. (2019); Bae et al. (2021)), which are effective in QSAR studies but limited by fixed designs and inherent bias to data-driven approaches. String-based representation like SMILES enables the application of NLP techniques to molecules. For example, Li et al. (2022a) use stacked CNN-RNN architectures, while Hou et al. (2022) and Segler et al. (2018) demonstrate that RNNs can capture SMILES syntax and the underlying chemical space distribution. Song et al. (2023) propose a double-head Transformer to extract detailed molecular features, and

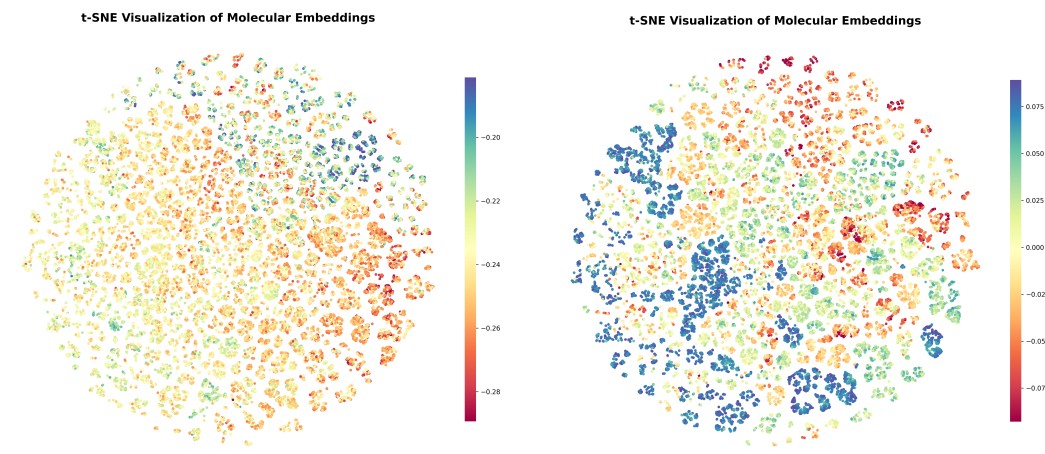

(a) Visualization of molecular representations learned by CLUSMOL by labeled with homo in qm9

(b) Visualization of molecular representations learned by CLUSMOL by labeled with lumo in qm9

Figure 6: Comparison of FGMG and similarity-based approaches.

Xie et al. (2025) enhance predictions by fusing domain knowledge with fine-tuned LLMs. Graph-based approaches using GNNs, such as AttentiveFP (Xiong et al. (2019)) and D-MPNN (Yang et al. (2019)), model molecules as graphs of atoms and bonds, achieving superior performance compared to traditional fingerprints. More recent works, including TopExpert (Kim et al. (2023)), use specialized models for molecules with similar topologies, while Chen et al. (2021) and Rollins et al. (2024) fuse multi-modal data (SMILES, 2D graphs, 3D structures) to further improve the quality of the representation.

**Self-supervised Learning in Chemistry.** SSL in chemistry includes generative, predictive, and contrastive approaches. Generative methods like VAEs (Hu et al. (2018); Samanta et al. (2020); Kingma et al. (2019)) reconstruct molecular data to learn meaningful embeddings. Predictive methods such as SMILES-BERT (Wang et al. (2019)) and ChemBERTa (Chithrananda et al. (2020)) use masked token prediction, while Hu et al. (2019) propose GNN pre-training at both node and graph levels. Motif-based SSL is explored in MGSSL (Zhang et al. (2021)), and HiMol (Zang et al. (2023)) combines generative and predictive tasks. Geometric information is emphasized in models such as GeM (Fang et al. (2022)) and Uni-Mol (Zhou et al. (2023)) for pre-training. Contrastive learning methods (Zhang et al. (2023)) have recently attracted significant attention. These approaches learn representations by maximizing agreement between different augmented views of the same molecule, while pushing apart representations of different molecules. GraphCL (You et al. (2020)) provides a foundational framework, improved by JOAO (You et al. (2021)) via automated augmentation. MolCLR (Wang et al. (2022)) and MoCL (Sun et al. (2021)) design molecule-specific augmentations. GraphMVP (Liu et al. (2021a)) and GenmGcl (Li et al. (2022b)) treat 2D-3D views as positive pairs, while 3DGCL (Moon et al. (2023)) contrasts 3D conformations from a low-energy pool. SMICLR (Pinheiro et al. (2022)) leverages SMILES and 2D graphs for accessible, effective representations.

## 6 CONCLUSION

In summary, the proposed CLUSMOL framework combines Function-Guided Graph Construction (FGMG) with Chemical Semantic-Weighted Contrastive Loss (SemCoL), enabling the model to learn more chemically meaningful molecular representations through pre-training. Experiments demonstrate that this approach not only outperforms existing methods across multiple downstream tasks but also validates the effectiveness of leveraging the intrinsic chemical community structure of molecules to guide feature learning. This opens new avenues for future research in molecular representation learning.

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

## USE OF LLM

Parts of this manuscript were drafted with the assistance of AI-based writing tools.

## A  PSEUDOCODE AND ALGORITHMS

### A.1  FGMG CONSTRUCTION ALGORITHM

---

**Algorithm 1** Functional Group-Guided Molecular Graphs (FGMG) Construction

---

**Require:** Molecular dataset $\mathcal{D} = \{m_1, \ldots, m_M\}$, Set of functional groups $\mathcal{F} = \{f_1, \ldots, f_K\}$
1: **Initialize:**
2:   Node set $\mathcal{V} \leftarrow \{m_1, \ldots, m_M\}$
3:   Edge set $\mathcal{E} \leftarrow \emptyset$
4:   Create a hash map canonical_smiles_to_idx from the canonical SMILES of each molecule in $\mathcal{D}$ to its index.
5:   **for** each molecule $m_i$ in $\mathcal{D}$ from $i = 1$ to $M$ **do**
6:     **for** each hydrogen atom $H$ in $m_i$ **do**
7:       **for** each functional group $f_k$ in $\mathcal{F}$ **do**
8:         Generate new molecule $m'_{\text{new}}$ by substituting $H$ in $m_i$ with $f_k$.
9:         Generate canonical SMILES $s'_{\text{new}}$ for $m'_{\text{new}}$.
10:        **if** $s'_{\text{new}}$ exists in canonical_smiles_to_idx **then**
11:          $j \leftarrow$ canonical_smiles_to_idx$[s'_{\text{new}}]$  ▷ Find the index of the resulting molecule.
12:          Add edge $(m_i, m_j)$ to $\mathcal{E}$, ensuring no duplicate or self-loops.
13:        **end if**
14:      **end for**
15:    **end for**
16:  **end for**
17:  **return** Graph $G = (\mathcal{V}, \mathcal{E})$

---

Let $M = |\mathcal{D}|$ denote the number of molecules, and $K = |\mathcal{F}|$ the number of functional groups. For each molecule $m_i \in \mathcal{D}$, let $H_i$ be the number of hydrogen atoms in $m_i$. We define $H = \max_{i=1}^{M} H_i$ as the maximum number of hydrogen atoms per molecule, and assume that the size of each molecule (e.g., number of atoms or bonds) is bounded by a constant $A$ relative to the input scale, as is common in molecular graph algorithms. Operations such as substituting a hydrogen with a functional group and computing the canonical SMILES string are performed using standard cheminformatics libraries (e.g., RDKit), which typically incur $O(A) = O(1)$ time per invocation under this bounded-size assumption, since $A$ is treated as constant for asymptotic analysis.

The algorithm constructs an undirected graph $G = (\mathcal{V}, \mathcal{E})$ where the vertices $\mathcal{V}$ correspond to the molecules in the dataset $\mathcal{D}$ of size $M$, and edges $\mathcal{E}$ connect pairs of molecules that differ by the substitution of a hydrogen atom with one of functional groups in $\mathcal{F}$. To analyze its time complexity, we consider the dominant operations and their frequencies, assuming standard computational models for molecular manipulations.

### A.1.1  INITIALIZATION PHASE

- Constructing the node set $\mathcal{V}$ requires $O(M)$ time, as it simply copies the dataset indices.
- Building the hash map canonical_smiles_to_idx involves computing the canonical SMILES for each of the $M$ molecules and inserting them into a hash table. Each canonicalization takes $O(1)$ time, and hash insertions are amortized $O(1)$ per operation, yielding a total of $O(M)$ time.

### A.1.2  MAIN LOOP PHASE

The core of the algorithm consists of three nested loops:

- The outer loop iterates over each of the $M$ molecules.

- For each molecule $m_i$, the middle loop iterates over its $H_i$ hydrogen atoms.

- For each hydrogen atom $h_i$, the inner loop iterates over the $K$ functional groups.

Within the innermost loop, the following operations occur:

1. Generate the new molecule $m'_{\text{new}}$ by substituting H with $f_k$. This is a local modification to the molecular structure, requiring $O(1)$ time.

2. Compute the canonical SMILES $s'_{\text{new}}$ for $m'_{\text{new}}$, again in $O(1)$ time.

3. Perform a hash map lookup for $s'_{\text{new}}$, which is $O(1)$ amortized time.

4. If the lookup succeeds, add the edge $(m_i, m_j)$ to $\mathcal{E}$, ensuring no duplicates or self-loops. Assuming $\mathcal{E}$ is implemented as a set of pairs (or an adjacency list with hash-based checks), this addition is $O(1)$ amortized.

The number of innermost iterations is $\sum_{i=1}^{M} H_i \cdot K \leq M \cdot H \cdot K$. Since each iteration performs $O(1)$ work, the main loop requires $O(MHK)$ time.

### A.1.3 OVERALL COMPLEXITY

Combining both phases, the total time complexity is $O(M) + O(MHK) = O(MHK)$.In fact, the total time complexity of the algorithm equals $O(M))$ as $H$ and $K$ are constants for each dataset $\mathcal{D}$ and $M \gg HK$.

### A.1.4 PROOF OF TIME COMPLEXITY

To prove this formally, we proceed by bounding the runtime step-by-step.

**Lemma 1:** The initialization phase runs in $O(M)$ time.
*Proof:* Computing canonical SMILES for $M$ molecules takes $O(M)$ time, and hash map construction is $O(M)$ amortized, as each insertion is independent.

**Lemma 2:** The main loop runs in $O(MHK)$ time.
*Proof:* The nested loops execute exactly $\sum_{i=1}^{M} H_i K$ iterations. By definition, $\sum_{i=1}^{M} H_i \leq MH$, so the iteration count is at most $MHK$. Each iteration performs a constant number of operations: substitution ($O(1)$), canonicalization ($O(1)$), lookup ($O(1)$), and conditional edge addition ($O(1)$). Thus, the total is $O(MHK)$.

**Theorem:** The algorithm has time complexity $O(MHK)$.
*Proof:* By Lemmas 1 and 2, the total runtime is $O(MHK)$ the sum of both phases.This bound is tight in the worst case, e.g., when every molecule has $H$ hydrogens, every substitution yields a distinct existing molecule, and no early terminations occur.To datasets $\mathcal{D}$ which $M \gg HK$,the time complexity simplified to $O(M)$.

### A.2 SEMCOL

The Chemical Semantic-Weighted Contrastive Loss is a contrastive loss function designed to align representations of molecules in a feature space, considering both molecule identity (same molecule) and label similarity (same label but different molecule). Below, we derive the loss function mathematically, explaining the key components and the role of the weighting mechanism.

**Problem Setup**

Given a batch of $B$ pairs of feature vectors $\mathbf{f}_1, \mathbf{f}_2 \in \mathbb{R}^{B \times D}$, corresponding labels $\mathbf{l}_1, \mathbf{l}_2 \in \mathbb{R}^B$, and molecule IDs $\mathbf{m}_1, \mathbf{m}_2 \in \mathbb{R}^B$, the goal is to compute a loss that encourages:

- Features of the same molecule (same $\mathbf{m}_i$) to be similar (weight 1).

- Features of different molecules with the same label (same $\mathbf{l}_i$, different $\mathbf{m}_i$) to be similar (weight $w_l$).

- Features of different molecules with different labels to be dissimilar.

---

**Algorithm 2** Chemical Semantic-Weighted Contrastive Loss

---

**Require:** Feature vectors $\mathbf{f}_1, \mathbf{f}_2 \in \mathbb{R}^{B \times D}$, labels $\mathbf{l}_1, \mathbf{l}_2 \in \mathbb{R}^B$, molecule IDs $\mathbf{m}_1, \mathbf{m}_2 \in \mathbb{R}^B$, temperature $\tau$, soft-positive samples weight $w_l$

1: Concatenate features: $\mathbf{f} \leftarrow [\mathbf{f}_1; \mathbf{f}_2]$                      ▷ [2B, D]
2: Concatenate labels: $\mathbf{l} \leftarrow [\mathbf{l}_1; \mathbf{l}_2]$                             ▷ [2B]
3: Concatenate molecule IDs: $\mathbf{m} \leftarrow [\mathbf{m}_1; \mathbf{m}_2]$              ▷ [2B]
4: Normalize features: $\mathbf{f} \leftarrow \mathbf{f}/\|\mathbf{f}\|_2$          ▷ Normalize along dimension 1
5: Compute similarity matrix: $\mathbf{S} \leftarrow (\mathbf{f} \cdot \mathbf{f}^\top)/\tau$
6: Compute maximum similarity: $\mathbf{s}_{\max} \leftarrow \max(\mathbf{S}, \dim = 1)$
7: Adjust similarities: $\mathbf{S} \leftarrow \mathbf{S} - \mathbf{s}_{\max}$            ▷ For numerical stability
8: Create self-exclusion mask: $\mathbf{M}_{\text{self}} \leftarrow (\text{eye}(2B) == 0)$     ▷ Exclude self-pairs
9: Create same-molecule mask: $\mathbf{M}_{\text{mol}} \leftarrow (\mathbf{m} = \mathbf{m}^\top) \wedge \mathbf{M}_{\text{self}}$
10: Create same-label mask: $\mathbf{M}_{\text{lab}} \leftarrow (\mathbf{l} = \mathbf{l}^\top) \wedge \mathbf{M}_{\text{self}} \wedge \neg\mathbf{M}_{\text{mol}}$
11: Compute positive mask: $\mathbf{M}_{\text{pos}} \leftarrow \mathbf{M}_{\text{mol}} + w_l \cdot \mathbf{M}_{\text{lab}}$
12: Compute exponential similarities: $\mathbf{E} \leftarrow \exp(\mathbf{S}) \cdot \mathbf{M}_{\text{self}}$
13: Compute numerator: $\text{num} \leftarrow \sum(\mathbf{E} \cdot \mathbf{M}_{\text{pos}}, \dim = 1)$
14: Compute denominator: $\text{den} \leftarrow \sum(\mathbf{E}, \dim = 1) + \epsilon$
15: Compute log probabilities: $\mathbf{p} \leftarrow \log(\text{num}/\text{den} + \epsilon)$
16: Create valid sample mask: $\mathbf{M}_{\text{valid}} \leftarrow \sum(\mathbf{M}_{\text{pos}}, \dim = 1) > 0$
17: Compute loss: $L \leftarrow -\text{mean}(\mathbf{p}[\mathbf{M}_{\text{valid}}])$
18: **return** $L$

---

The features are concatenated into $\mathbf{f} = [\mathbf{f}_1; \mathbf{f}_2] \in \mathbb{R}^{2B \times D}$, labels into $\mathbf{l} = [\mathbf{l}_1; \mathbf{l}_2] \in \mathbb{R}^{2B}$, and molecule IDs into $\mathbf{m} = [\mathbf{m}_1; \mathbf{m}_2] \in \mathbb{R}^{2B}$.

**Step 1: Feature Normalization**

The features are normalized to unit length:

$$\mathbf{f}_i \leftarrow \frac{\mathbf{f}_i}{\|\mathbf{f}_i\|_2}, \quad i = 1, \dots, 2B,$$

ensuring that similarity comparisons are based on cosine similarity (dot product of normalized vectors).

**Step 2: Similarity Matrix**

The similarity matrix is computed as:

$$\mathbf{S}_{ij} = \frac{\mathbf{f}_i \cdot \mathbf{f}_j}{\tau}, \quad i, j = 1, \dots, 2B,$$

where $\tau > 0$ is the temperature parameter scaling the similarities. For numerical stability, the maximum similarity per row is subtracted:

$$\mathbf{S}_{ij} \leftarrow \mathbf{S}_{ij} - \max_j \mathbf{S}_{ij}.$$

This prevents overflow in the exponential terms, as:

$$\exp(\mathbf{S}_{ij} - \max_j \mathbf{S}_{ij}) = \exp(\mathbf{S}_{ij})/\exp(\max_j \mathbf{S}_{ij})$$

.

**Step 3: Positive and Negative Samples**

Define the following masks:

- **Self-exclusion mask**: $\mathbf{M}_{\text{self},ij} = 1$ if $i \neq j$, else 0 (excludes self-similarities).

- **Same-molecule mask**: $\mathbf{M}_{\text{mol},ij} = 1$ if $\mathbf{m}_i = \mathbf{m}_j$ and $i \neq j$, else 0.

- **Same-label mask**: $\mathbf{M}_{\text{lab},ij} = 1$ if $\mathbf{l}_i = \mathbf{l}_j$, $\mathbf{m}_i \neq \mathbf{m}_j$, and $i \neq j$, else 0.

The positive mask combines these with weights:

$$\mathbf{M}_{\text{pos},ij} = \mathbf{M}_{\text{mol},ij} + w_l \cdot \mathbf{M}_{\text{lab},ij},$$

where $w_l \in [0, 1]$ (e.g., 0.3) reduces the contribution of same-label pairs compared to same-molecule pairs.

**Step 4: Contrastive Loss**

The contrastive loss is based on the InfoNCE loss, adapted for weighted positive samples. For each anchor feature $\mathbf{f}_i$, the loss encourages high similarity with positive samples (defined by $\mathbf{M}_{\text{pos}}$) relative to all other samples. Compute:

$$\mathbf{E}_{ij} = \exp(\mathbf{S}_{ij}) \cdot \mathbf{M}_{\text{self},ij},$$

where $\mathbf{E}_{ij}$ is the exponentiated similarity for non-self pairs. The numerator sums the contributions of positive samples:

$$\text{num}_i = \sum_{j=1}^{2B} \mathbf{E}_{ij} \cdot \mathbf{M}_{\text{pos},ij} = \sum_{j:\mathbf{M}_{\text{pos},ij}>0} \exp\left(\frac{\mathbf{f}_i \cdot \mathbf{f}_j}{\tau}\right) \cdot \mathbf{M}_{\text{pos},ij},$$

where $\mathbf{M}_{\text{pos},ij} \in \{1, w_l\}$ applies the appropriate weight. The denominator sums over all non-self pairs:

$$\text{den}_i = \sum_{j=1}^{2B} \mathbf{E}_{ij} = \sum_{j \neq i} \exp\left(\frac{\mathbf{f}_i \cdot \mathbf{f}_j}{\tau}\right) + \epsilon,$$

with $\epsilon = 10^{-9}$ for numerical stability. The log probability for anchor $i$ is:

$$p_i = \log\left(\frac{\text{num}_i}{\text{den}_i} + \epsilon\right).$$

Only anchors with at least one positive sample ($\sum_j \mathbf{M}_{\text{pos},ij} > 0$) contribute to the loss:

$$L = -\frac{1}{|\{i : \sum_j \mathbf{M}_{\text{pos},ij} > 0\}|} \sum_{i:\sum_j \mathbf{M}_{\text{pos},ij}>0} p_i.$$

**Interpretation**

The loss minimizes the negative log probability of positive pairs being similar, weighted by 1 for same-molecule pairs and $w_l$ for same-label pairs. This encourages the model to learn representations where same-molecule pairs are closer in the feature space than same-label pairs, which are in turn closer than negative pairs (different labels and molecules). The temperature $\tau$ controls the softness of the distribution, and $w_l$ balances the influence of label-based positives versus molecule-based positives.

This formulation extends the standard contrastive loss by incorporating a weighted positive mask, allowing flexible handling of domain-specific relationships (molecule identity and label similarity) in molecular representation learning.

## B  DETAIL OF DOWNSTREAM DATASET

Table 3: Detailed Statistics of MoleculeNet Downstream Tasks

| Category | Dataset | Data Type | Task Type | # Tasks | # Compounds | Rec. Split | Rec. Metric |
|---|---|---|---|---|---|---|---|
| Quantum Mechanics | QM8 | SMILES | Regression | 12 | 21786 | Random | MAE |
| Quantum Mechanics | QM9 | SMILES | Regression | 12 | 133885 | Random | MAE |
| Physical Chemistry | ESOL | SMILES | Regression | 1 | 1128 | Random | RMSE |
| Physical Chemistry | FreeSolv | SMILES | Regression | 1 | 642 | Random | RMSE |
| Physical Chemistry | Lipo | SMILES | Regression | 1 | 4200 | Random | RMSE |
| Physiology | BACE | SMILES | Classification | 1 | 1513 | Scaffold | ROC-AUC |
| Physiology | BBBP | SMILES | Classification | 1 | 2039 | Scaffold | ROC-AUC |
| Physiology | Tox21 | SMILES | Classification | 12 | 7831 | Random | ROC-AUC |
| Physiology | SIDER | SMILES | Classification | 27 | 1427 | Random | ROC-AUC |
| Physiology | ClinTox | SMILES | Classification | 2 | 1478 | Random | ROC-AUC |

In this section, we review the main categories of datasets used for downstream tasks.

### B.1 Physiology

The Blood-Brain Barrier Penetration (BBBP) dataset measures whether a molecule will penetrate the central nervous system. All three datasets, Tox21, ToxCast, and ClinTox, are related to the toxicity of molecular compounds. The Side Effect Resource (SIDER) dataset stores the adverse drug reactions on a marketed drug database. BACE measures the binding results for a set of inhibitors of $\beta$-secretase 1 (BACE-1).

- **BBBP** (Martins et al. (2012)) assesses the ability of a compound to cross the blood-brain barrier, a critical factor in drug design for central nervous system disorders. The dataset contains 2,039 compounds with binary labels indicating permeability.
- **BACE** (Subramanian et al. (2016)) contains 1,513 compounds with binding affinity data for BACE-1 inhibitors, available in both classification and regression formats.
- **Tox21** (Huang et al. (2016)) is part of the Toxicology in the 21st Century initiative, evaluating the toxicity of 7,831 compounds across 12 tasks. The dataset includes sparse multi-label annotations for various toxic endpoints.
- **ClinTox** (Gayvert et al. (2016)) compares the toxicity profiles of 1,478 FDA-approved drugs against those that failed clinical trials, with two binary tasks.
- **SIDER** (Kuhn et al. (2016)) documents adverse drug reactions for 1,427 marketed drugs across 27 side effect categories.

### B.2 Physical Chemistry

Dataset proposed in (Delaney (2004)) measures aqueous solubility of the molecular compounds. Lipophilicity (Lipo) dataset (Wu et al. (2018)) is a subset of ChEMBL (Gaulton et al. (2012)) measuring the molecule octanol/water distribution coefficient.

- **ESOL** (Delaney (2004)) quantifies the aqueous solubility (log mol/L) of 1,128 compounds, a key property for drug formulation..
- **FreeSolv** (Mobley & Guthrie (2014)) provides hydration free energy (kcal/mol) for 642 small molecules, aiding in solvation studies..
- **Lipo** (Wu et al. (2018)) is a subset of ChEMBL (Gaulton et al. (2012)) with 4,200 compounds, measuring lipophilicity (logD) to assess compound behavior in biological systems.

### B.3 Quantum Mechanics

QM8 and QM9 datasets contain quantum mechanical properties derived from density functional theory (DFT), providing large-scale resources for molecular representation learning.

- **QM8** (Öztürk et al. (2018)) contains 21,786 molecules with 12 quantum mechanical properties (e.g., electronic energies), derived from density functional theory (DFT).
- **QM9** (Öztürk et al. (2018)) extends this with 133,885 molecules and 13 properties (e.g., HOMO/LUMO, dipole moment), offering a large-scale quantum dataset.

## C Graph-Label Dataset Details

In this section, we provide a detailed explanation of the statistical information obtained from the molecular graphs generated by the FGMG algorithm in our experiments, the set of functional groups selected for the experiments, and the statistical information of the clustering results. First, we employed the functional group set shown in Table 4, which encompasses most functional groups critical to molecular properties to ensure the efficacy of the constructed molecular graph. Statistical information from the molecular graph generated using the FGMG algorithm is presented in the Figure 7. Subsequently, we applied the Louvain algorithm to the network, yielding the clustering results depicted in Figure 8. Statistical analysis indicates that the majority of edges between molecules belong to the Alkanes and Oxygen functional groups. The clustering results reveal that most communities fall within the size range of 0–100, with communities exceeding 10,000 in size constituting a very small proportion.

Table 4: Functional Group SMILES Representations

| Alkanes/Alkyl | Alkenes | Alkynes | Aromatics | Halogens | Oxygen | Nitrogen |
|---|---|---|---|---|---|---|
| [*]C | [*]C=C | [*]C#C | [*]c1ccccc1 | [*]F | [*]O | [*]N |
| [*]CC | [*]CC=C | [*]CC#C | [*]c1ccncc1 [*]Cl | [*]C=O | [*]C#N | [*]C(=O)S |
| [*]CCC | [*]C=CC | [*]C#CC | [*]c1ccc(cc1)F | [*]Br | [*]C(=O)O | [*]N=C=O |
| [*]CCCC | [*]C=C=C | [*]C=CC#C | [*]c1ccc(cc1)Cl | [*]I | [*]C(=O)Cl | [*]NC=O |
| [*]C1CC1 | [*]C=CCC | | [*]c1ccc(cc1)Br | [*]C=C(Cl) | [*]C(=O)C | [*]NC(=O)O |
| [*]C1CCC1 | [*]C=CC=C | | [*]c1ccc(cc1)I | [*]C=C(F)C | [*]C(=O)OC | [*]NCO |
| [*]C1CCCC1 | | | | [*]C(Cl)(Cl)Cl | [*]CO | [*]CN |
| [*]C1CCCCC1 | | | | [*]C(F)(F)F | [*]OC | [*]CCN |

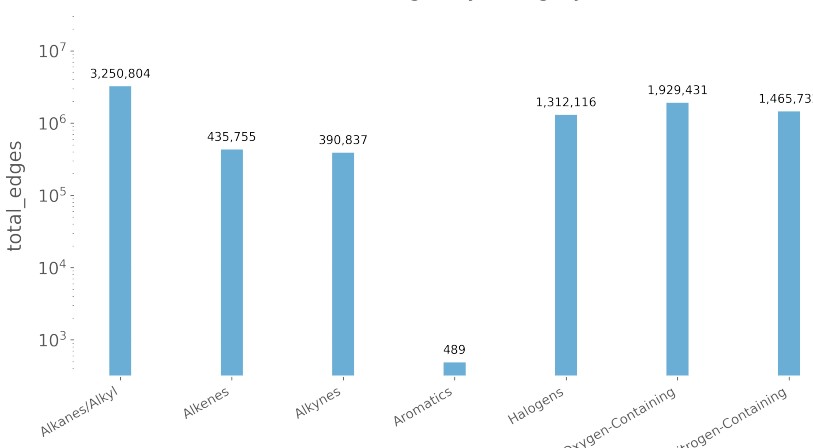

Figure 7: Molecular Graph Statistics

# D EXPERIMENT

## D.1 EXPERIMENT SETTING

In the pre-training phase of CLUSMOL, we adopt a carefully tuned configuration to optimize the cluster-assisted contrastive learning framework. The learning rate is set to $5 \times 10^{-4}$, with a weight decay of $1 \times 10^{-5}$ to regularize the model. The SemCoL weight is configured at 0.5, and the temperature parameter is fixed at 0.07 to balance the contribution of positive and negative pairs in the contrastive loss. A batch size of 256 is employed, and the model is trained for 100 epochs. The GNN encoder's output is processed through a single-layer MLP with a linear transformation to compute the contrastive loss, and model parameters are updated using the Adam optimizer. For data augmentation, we apply an edge deletion rate of 0.25 and a node masking rate of 0.2.

For downstream tasks, we split the datasets using an 8:1:1 ratio for training, validation, and test sets, respectively, ensuring a robust evaluation of the pre-trained representations. The batch size is reduced to 64, and the learning rate is maintained at $5 \times 10^{-4}$.

## D.2 BASELINE

### SUPERVISED BASELINES

**Attentive FP**: A graph attention network for molecular graphs, using attention mechanisms to weigh atom and bond contributions for property prediction. It excels in tasks like solubility and toxicity prediction.

**D-MPNN**: A directed message-passing neural network that aggregates edge features to model molecular graphs. It improves efficiency by preventing redundant messaging and is used for QSPR and hERG prediction.

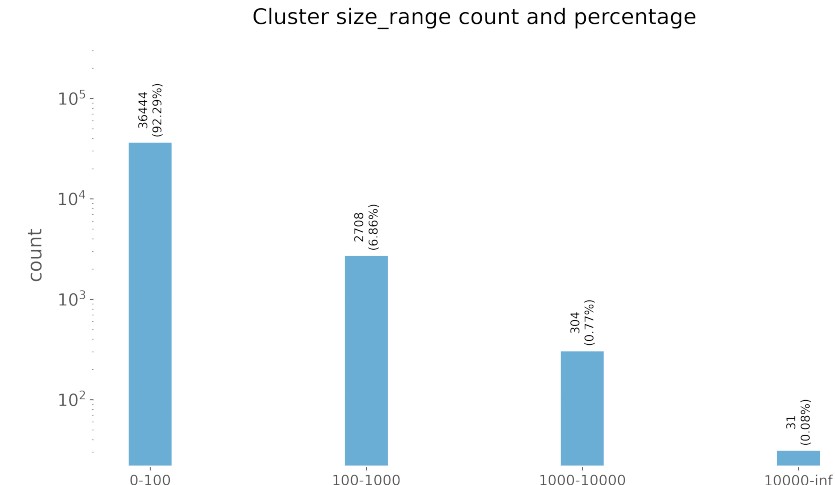

Figure 8: Number of communities in different-sized molecular communities

SELF-SUPERVISED BASELINES: GENERATIVE APPROACHES

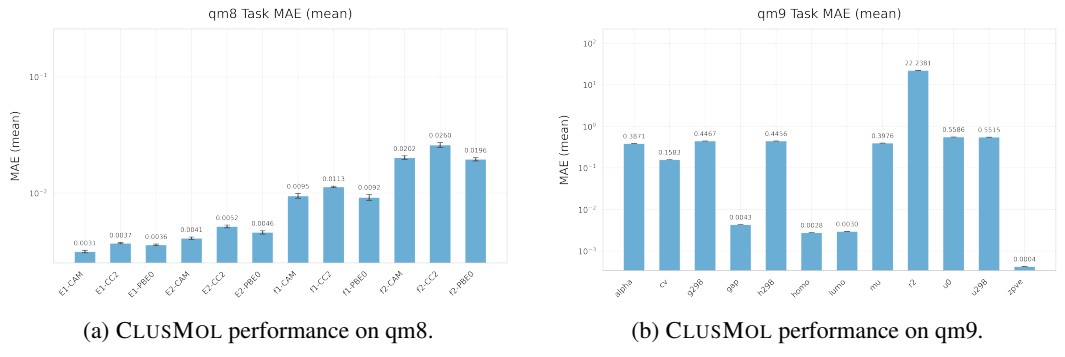

(a) CLUSMOL performance on qm8.  (b) CLUSMOL performance on qm9.

Figure 9: Comparison of FGMG and similarity-based approaches.

**InfoGraph**: Maximizes mutual information between graph and substructure representations using a contrastive loss. It enables robust pretraining for graph classification tasks.

**AttrMask**: Trains GNNs to reconstruct randomly masked attribute in molecular graphs. It captures local topology for downstream tasks like link prediction.

**ContextPred**: Predicts contextual relationships between subgraphs in a graph using contrastive learning. It learns structural patterns for molecular property prediction.

**KGG**: The KGG framework comprises two primary modules: the Knowledge Representation Graph (KRG) serves as an encoder, extracting a hierarchical graph representation enhanced by knowledge tracks; Knowledge Self-Supervised Pretraining (KSSP) functions as a multi-task pretraining decoder, reconstructing knowledge from the KRG representation using an MLP.

SELF-SUPERVISED BASELINES: CONTRASTIVE METHODS

**GraphCL**: Uses graph augmentations and contrastive learning to learn invariant representations. It supports node and graph classification in molecular tasks.

**GraphMVP**: Combines 2D and 3D molecular views via contrastive and generative pretraining. It enhances representations for drug discovery applications.

**MolCLR**: Applies contrastive learning with augmentations like atom masking to learn molecular representations. It transfers well to property prediction tasks.

### D.3 CLUSMOL RESULTS DETAIL

Figure 10: Comparison of Morgan Fingerprint and CLUSMOL Metrics on qm8

In Figure 9a and Figure 9b, we present detailed test results for CLUSMOL on the qm8 and qm9 datasets. As shown, CLUSMOL achieves excellent performance across all tasks in both qm8 and qm9. Additionally, Figure 10compares CLUSMOL and Morgan Fingerprint across qm8 tasks under the constraint of frozen the weights of CLUSMOL encoder. It is evident that for most qm8 tasks, fingerprints trained by CLUSMOL outperform those generated by Morgan Fingerprint.

