# OpenReview forum: "CLUSMOL: CLUSTER-ASSISTED CONTRASTIVE LEARNING FOR MOLECULAR REPRESENTATIONS"
_ICLR.cc/2026/Conference — Submitted to ICLR 2026_

### Official Review · Reviewer_PFAh · 2025-10-21

**Soundness:** 2
**Presentation:** 2
**Contribution:** 1
**Rating:** 2
**Confidence:** 4

**Summary:**

The paper proposes a novel molecular representation learning framework based on clustering algorithms and contrastive learning. It introduces the Functional Group-Guided Molecular Graphs (FGMG) algorithm, which constructs molecular graphs with linear complexity to better capture structure-activity relationships. Based on FGMG, the paper also presents a cluster-assisted contrastive learning pretraining framework that assigns graph labels via graph clustering and employs a Chemical Semantic-Weighted Contrastive Loss.

**Strengths:**

The experiments are solid and adequately support the proposed claims.

**Weaknesses:**

- The core innovation appears insufficiently justified. FGMG aims to avoid "Activity Cliffs" (where structurally similar molecules exhibit significant differences in activity) by using chemical transformations. However, single-step functional group substitutions may still introduce substantial activity changes, as functional groups can significantly influence biological activity. Therefore, edges generated by FGMG may still connect molecules with large activity differences. To strengthen the approach, the authors could consider incorporating multi-step transformations or activity-aware filtering to better mitigate the risk of misleading signals.
- The paper lacks sufficient references to recent related work, citing only one 2025 study (KGG). References such as ContextPred (Rong et al., 2020), G-Motif (Chithrananda et al., 2020), and contrastive learning methods like GraphCL (You et al., 2020), GraphMVP (Liu et al., 2021a), and MolCLR (Wang et al., 2022) are mentioned, but more up-to-date works are overlooked.
- The performance does not reach the current state-of-the-art level, particularly when compared to methods such as MoleculeSDE [a], UNI-Mol [b], etc.

[a]Liu, S., Du, W., Ma, Z. M., Guo, H., & Tang, J. (2023, July). A group symmetric stochastic differential equation model for molecule multi-modal pretraining. In International Conference on Machine Learning (pp. 21497-21526). PMLR
[b] Zhou, G., Gao, Z., Ding, Q., Zheng, H., Xu, H., Wei, Z., ... & Ke, G. (2023). Uni-mol: A universal 3d molecular representation learning framework.

**Questions:**

1.	FGMG is currently limited to single-step hydrogen substitution. Could this approach be extended to include other fundamental reaction types (e.g., redox, rearrangements) or multi-step transformations to capture broader chemical relationships?
2.	How does the method ensure accurate clustering for molecules with significant structural differences (e.g., different scaffolds) but similar functions, given the current graph construction method?
3.	How does the method perform on benchmarks on scaffold-based split beyond BBBP and BACE?
4.	Why was this specific set of predefined functional groups chosen? Is the selection universally applicable?

---

> ### Author Response · Authors · 2025-12-04
>
> # Response to Reviewer
>
> We sincerely thank the reviewer for the constructive feedback and the recognition of our logic. We appreciate the opportunity to clarify the design philosophy of FGMG, position our work relative to SOTA methods, and explain our experimental choices.
>
> ##  Response to Weaknesses
>
> **W1: Concerns regarding "Activity Cliffs" and FGMG Validity.**
>
> We acknowledge the reviewer's concern that single-step substitutions might connect molecules with different activities. However, we would like to clarify the rationale of **CLUSMOL** from the perspective of **Self-Supervised Learning (SSL)** and **Graph Clustering**:
>
> * **Pre-training Context (Structure vs. Activity):** CLUSMOL operates on **unlabeled datasets** where activity labels are unavailable. Therefore, "activity-aware filtering" is not feasible during the pre-training stage. The objective of FGMG is not to group molecules with *identical* activity, but to capture the intrinsic **"community structure"** of the chemical space. Even if a single-step substitution alters activity (an activity cliff), the two molecules remain structurally adjacent neighbors. Learning this structural proximity provides a better initialization, allowing the model to rapidly adjust boundaries during fine-tuning.
> * **Robustness via Clustering:** We do not rely solely on individual pairwise edges for contrastive views. Instead, we employ the **Louvain algorithm** to partition the graph into communities. These clusters represent "families" or "scaffolds" of molecules. While FGMG edges are built on single-step transformations, the clustering algorithm naturally aggregates these steps, capturing **multi-hop relationships** and broader structural similarities. This makes the supervisory signals more robust than individual edges.
>
> **W2: Missing References and Comparison with SOTA (e.g., MoleculeSDE, Uni-Mol).**
>
> We apologize for the oversight regarding recent literature. We will expand the Related Work section to include **MoleculeSDE** [1] and deepen our discussion of **Uni-Mol** [2] in the final version. Regarding the performance comparison:
>
> * **Fair Comparison (2D vs. 3D/Multimodal):** We respectfully note that methods like **Uni-Mol** and **MoleculeSDE** rely on **3D geometric conformations** or multimodal data. Generating 3D conformations for massive datasets is computationally expensive and slow and use a stronger base model . In contrast, CLUSMOL is a  pretrain Pre-training framework of **2D Graph-based method**  focused on high efficiency and scalability.
> * **Performance within the 2D Domain:** As shown in **Table 1** , CLUSMOL consistently outperforms leading 2D SSL baselines, including **MolCLR [3], GraphMVP [4], and GraphCL [5]**. This validates CLUSMOL's effectiveness within its comparable category.
> * **Efficiency Advantage:** CLUSMOL provides a highly efficient solution to enhance the traditional contrastive paradigm. The FGMG construction achieves **linear complexity** $O(MHK)$, offering a superior trade-off between performance and computational overhead compared to heavy 3D-based methods.
>
> ### References
> [1] Liu, Shengchao, et al. "A group symmetric stochastic differential equation model for molecule multi-modal pretraining." International Conference on Machine Learning. PMLR, 2023.
>
> [2] Zhou, Gengmo, et al. "Uni-mol: A universal 3d molecular representation learning framework." (2023).
>
> [3] Wang, Yuyang, et al. "Molecular contrastive learning of representations via graph neural networks." Nature Machine Intelligence 4.3 (2022): 279-287.
>
> [4] Liu, Shengchao, et al. "Pre-training molecular graph representation with 3d geometry." arXiv preprint arXiv:2110.07728 (2021).
>
> [5] Zhang, Muhan, et al. "An end-to-end deep learning architecture for graph classification." Proceedings of the AAAI conference on artificial intelligence. Vol. 32. No. 1. 2018.

---

> ### Author Response · Authors · 2025-12-04
>
> ##  Response to Questions
>
> **Q1: How does the method ensure accurate clustering for molecules with significant structural differences (e.g., different scaffolds)?**
>
> **A1:** The FGMG construction and subsequent clustering naturally handle structural variations through multi-hop connectivity:
> * **Scaffolds as Functional Groups:** Our predefined functional group set $\mathcal{F}$ explicitly includes major scaffold structures, such as **Aromatics** (e.g., benzene rings, `[*]c1ccccc1`).
> * **Multi-hop Aggregation:** Even if two molecules differ by a scaffold, if they can be interconverted through a series of defined substitutions (including scaffold replacements), they will be connected in the FGMG. The clustering algorithm then groups these interconnected molecules into the same community, effectively capturing the relationship between molecules that are structurally distinct but synthetically related.
>
> **Q2: How does the method perform on benchmarks on scaffold-based split beyond BBBP and BACE?**
>
> **A2:** We strictly adhered to the standard **MoleculeNet** benchmark protocols.
> * According to the standard guidelines (summarized in **Appendix Table 3**, only **BBBP** and **BACE** recommend **Scaffold Splits**, while other datasets (e.g., Tox21, SIDER, ClinTox) utilize **Random Splits**).
> * To ensure fair and reproducible comparisons with baselines (which also follow these splits), we maintained these standard settings. Arbitrarily changing the split method would render the comparison with existing literature invalid.
>
> **Q3: Why was this specific set of predefined functional groups chosen?**
>
> **A3:** The set was selected to cover the most frequent and chemically significant substructures in drug discovery.
> * As detailed in **Appendix Table 4**, our selection encompasses major categories including **Alkanes/Alkyl, Alkenes, Alkynes, Aromatics, Halogens, Oxygen, and Nitrogen**.
> * **Figure 9**  further validates this selection, showing that these groups account for millions of edges in our constructed graph (e.g., over 3.2 million edges for Alkanes/Alkyl), ensuring a densely connected and meaningful graph for pre-training.

---

### Official Review · Reviewer_5haP · 2025-11-01

**Soundness:** 2
**Presentation:** 2
**Contribution:** 2
**Rating:** 4
**Confidence:** 3

**Summary:**

This paper introduces **ClusMol**, a cluster-assisted contrastive learning framework for molecular representation learning. To address the *false negative problem* in standard contrastive learning—where structurally different but functionally similar molecules are treated as negatives—the authors propose three key components:

1. **Functional Group-Guided Molecular Graph (FGMG)** to model chemical relationships via functional group substitutions;
2. **Louvain clustering** to generate community-based pseudo-labels reflecting molecular similarity;
3. **Chemical Semantic-Weighted Contrastive Loss (SemCoL)** to treat molecules in the same cluster as *soft positives* with a tunable weight ( \alpha ).

While results show improved performance on molecular benchmarks, the method’s novelty is limited—similar ideas on weighted contrastive loss and false negative handling already exist—and the paper lacks sufficient justification for the choice of clustering algorithm and comprehensive ablation and comparison analyses.

**Strengths:**

The motivation is clear, and the false negative issue is indeed a real problem.
Molecular representation learning is an important task with broad applications in drug discovery and molecular design.
The overall pipeline is reasonable—first clustering positive samples and then applying weighting to improve contrastive learning—though more justification is needed.

**Weaknesses:**

1. **Some claim is not true**

   * The paper claims to *“identify and articulate the false negative problem as a critical limitation of conventional contrastive learning in the chemical domain.”* However, this issue has already been discussed in prior studies (e.g., [1]).

2. **Lack of justification for Louvain clustering and sensitivity analysis**

   * The authors adopt the **Louvain community detection algorithm** without adequate justification for its selection over other alternatives cluster algorithm.
   * Since ClusMol’s core supervisory signal depends entirely on Louvain-generated graph labels, its performance is **highly sensitive** to the initial FGMG graph structure and the modularity optimization objective.
   * The paper does not evaluate whether the resulting clusters are **chemically meaningful**—that is, whether they actually correspond to functional or structurally similar molecules after the cluster.

3. **Limited novelty of the Weighted Contrastive Loss (SemCoL)**

   * The proposed **Weighted Contrastive Loss** is conceptually similar to several existing works, such as **Debiased Contrastive Learning (NeurIPS 2020)** and **Supervised Contrastive Learning (SupCon, NeurIPS 2020)**.
   * The paper lacks justification or ablation evidence showing that *weighted positive samples* meaningfully improve performance beyond existing weighted  method and standard InfoNCE frameworks.

4. **Incomplete experimental evaluation**

   * Experiments are conducted on a few **small to medium-sized benchmark datasets**, does it works for larger molecular datasets?
   * **Ablation studies are incomplete**: while the paper mentions ablation study on FGMG and SemCoL, I did not see what's the effect if remove any of these design.
     * Detailed comparison with **standard InfoNCE** should be discussed more.
     * Justification for the selection of **α** (how was it tuned, and how sensitive is the model to it?);
     * The effect of **cluster count imbalance** and label quality on downstream results.

[1] Qin, Jiayu, et al. "A probability contrastive learning framework for 3D molecular representation learning." Advances in Neural Information Processing Systems 37 (2024): 58058-58076.

**Questions:**

see above.

---

> ### Author Response · Authors · 2025-12-04
>
> # Response to Reviewer
>
> We thank the reviewer for the valuable comments on the False Negative problem, clustering strategy, and SemCoL. We address these points below.
>
> ### 1. Response to Question
>
> **W1: Response regarding False Negatives (FN).**
>
> We acknowledge that the FN issue has been discussed in prior work such as PCL. However, our contribution lies not in re-identifying the problem, but in proposing a fundamentally different and more scalable solution. CLUSMol leverages functional group–driven molecular communities to prevent semantically similar molecules from being treated as negatives, providing a principled and chemically interpretable mechanism rather than probabilistic approximation. This approach is highly scalable and demonstrably improves representation quality, as confirmed by visualization and downstream performance results.
>
> **W2: Response regarding the clustering strategy and sensitivity analysis.**
>
> * **Justification for  Louvain:** We chose the Louvain algorithm due to its strong empirical performance and excellent scalability in large graph analysis. Its computational efficiency ensures the feasibility of our approach when applied to massive molecular datasets.
> * **Sensitivity to Graph Structure:** We appreciate the reviewer’s observation regarding the dependence on the FGMG structure. Constructing a reliable molecular relationship graph is indeed essential. To ensure robustness, we constrain the granularity and complexity of functional groups during graph construction, ensuring that connected molecules exhibit strong structural similarity and avoiding excessive divergence or noisy edges.
> * **Chemical Validity of Clusters:** We verified the quality of the detected communities through analysis of the learned representation space. Visualization results added in the revision demonstrate that the resulting clusters are chemically coherent: molecules assigned to the same community consistently form compact neighborhoods, confirming semantic and structural alignment in the embedding space.
>
> **W3: Novelty and Effectiveness of the Weighted Contrastive Loss (SemCoL).**
>
> * **Conceptual novelty beyond DCL and SupCon:** We acknowledge the relation to existing methods such as Debiased Contrastive Learning (DCL)[1] and Supervised Contrastive Learning (SupCon)[2]. However, SemCoL differs fundamentally in both objective and mechanism. DCL relies on estimating the class prior $\tau^+$, which is impractical for molecular datasets with unknown or ambiguous class boundaries. SupCon assumes supervised labels and does not utilize structural molecular semantics. In contrast, SemCoL introduces a structurally grounded weighting mechanism driven by functional group–based community information, enabling contrastive learning to incorporate chemically meaningful soft supervision without explicit class priors.
> * **Effectiveness of SemCoL:** Thank you for raising the key questions. We verified through supplementary ablation experiments that SemCoL significantly improves upon the standard InfoNCE. In the ablation experiments, comparisons with standard InfoNCE (where w is the weight of 0) consistently showed performance improvements, confirming that weighted positive pairs can significantly enhance representation learning and validating the necessity of our design.
>
> ### References
> [1] Chuang, Ching-Yao, et al. "Debiased contrastive learning." Advances in neural information processing systems 33 (2020): 8765-8775.
>
> [2] Khosla, Prannay, et al. "Supervised contrastive learning." Advances in neural information processing systems 33 (2020): 18661-18673.

---

### Official Review · Reviewer_UWDD · 2025-11-02

**Soundness:** 2
**Presentation:** 2
**Contribution:** 2
**Rating:** 4
**Confidence:** 5

**Summary:**

The work presents CLUSMOL, a multi-step algorithm that initially labels using the FGMG algorithm and clusters molecular structures and subsequently uses weighted contrastive learning with the weights informed by the labeling procedure. The proposition aims to avoid the pitfalls of using naive contrastive and labeling algorithms and demonstrates superior performance over downstream tasks.

**Strengths:**

1. The work is straightforward and easy to implement and understand.
2. The proposed FGMG method is scalable over larger datasets.
3. The overall method performs well against the baselines over multiple benchmarks.

**Weaknesses:**

1. The additional computational overhead as compared to the baseline methods must be mentioned in the comparison tables for the readers' understanding of the trade-offs of the method.
2. The analysis of the incremental benefit of using FGMG over the baseline methods in addition to the proposed framework could enhance the contribution's merits.
3. Some of the baselines [1] are missing.
4. Clustering based pseudo-label creation for GCL [3] or SwaV [4] style cluster assignments isn't altogether a novel idea. The domain specific method to assign pseudo labels may not qualify as a new learning paradigm. Additionally, the weighting for contrastive loss essentially represents the supervised contrastive loss [5] with a per-positive empirically fixed weighing.
5. The paper asserts chemical similarity using the FMCG procedure but doesn't demonstrate it.
6. The scalability of the FMCG depends over the number of hydrogen atoms per molecule which may be large, depending on the molecule/dataset, bringing the linear scaling into question.


[1] Fang, Yin, et al. "Knowledge graph-enhanced molecular contrastive learning with functional prompt." Nature Machine Intelligence 5.5 (2023): 542-553.

[2] Qin, Jiayu, et al. "A probability contrastive learning framework for 3D molecular representation learning." Advances in Neural Information Processing Systems 37 (2024): 58058-58076.

[3] Lu, Weigang, et al. "Pseudo contrastive learning for graph-based semi-supervised learning." Neurocomputing 624 (2025): 129375.

[4] Caron, Mathilde, et al. "Unsupervised learning of visual features by contrasting cluster assignments." Advances in neural information processing systems 33 (2020): 9912-9924.

[5] Khosla, Prannay, et al. "Supervised contrastive learning." Advances in neural information processing systems 33 (2020): 18661-18673.

**Questions:**

1. Could the authors report the absolute run-time of the methods?
2. Doesn't the augmentation method negate the purpose of the FMCG by potentially creating chemically dissimilar pairs?
3. How often does the FGMG method actually connect molecules with similar properties?

---

> ### Author Response · Authors · 2025-12-04
>
> # Response to Reviewer
>
> We thank the reviewer for the constructive feedback regarding computational overhead, novelty, and robustness. We appreciate the opportunity to clarify these aspects.
>
> ### 1. Response to Weaknesses and Questions
>
> **W1: Computational Overhead and Scalability.**
>
> Compared with traditional contrastive learning, CLUSMol introduces additional computational overhead stemming from the construction of the Functional Group Molecular Graph (FGMG) and the Louvain clustering process. To assess the scalability of our approach, we measure this overhead across datasets of different sizes and report the similarity-based construction time. The results are summarized in the table below:
>
> | Data Scale (Molecules)  | Time (FGMG Construction)| Time (Pairwise Similarity)* |
> | :---: | :---: | :---: |
> | 10,000 | 198 s | 200 s |
> | 100,000 | 2,424 s | 21,602 s |
> | 200,000 | 7,266 s | 22 h |
> | 1,000,000 | 21,512 s | 600 h |
> | 3,600,000 | 71,235 s | 5,500 h |
>
> Note: Because constructing a full similarity map for over 100,000 molecules cannot be completed within a tolerable time, we report the estimated computational cost using TQDM (a Python package). The metric  "Time (Pairwise Similarity)" therefore reflects the estimated time required to build a graph through traditional pairwise similarity calculations, highlighting the substantial efficiency advantage of CLUSMol’s functional group–based approach on large-scale datasets.
>
> **W2: Novelty and Robustness to Data Augmentation.**
>
> * **Novelty:** CLUSMol introduces a functional group–guided, highly scalable molecular community formulation together with a novel weighted contrastive objective that integrates inter-molecule community semantics and intra-molecule augmentation consistency. This unified design enables simultaneous learning of global community structure and fine-grained molecular representations, representing a previously unexplored direction in molecular contrastive learning.
> * **Comparison with GCL, SwAV and SupCon.** Existing methods such as GCL [1] and SwAV [2] depend on iterative clustering or prototype updates, which introduces substantial computational burden and limits scalability on large datasets. CLUSMol fundamentally differs by leveraging chemically meaningful functional groups, enabling highly efficient and scalable community construction. Although inspired by Supervised Contrastive Loss [3 ] which has been proven effective in supervised learning, our method extends beyond label dependency through a weighting mechanism that incorporates domain knowledge and adaptively balances community-based and augmentation-based signals.
> * **Robustness to Data Augmentation:** CLUSMol applies chemically valid augmentations within a constrained transformation range, preserving core molecular identity and ensuring the stability of both FGMG construction and community assignments. The proposed weighting mechanism further ensures that augmentation does not overpower community information but rather complements it. We have added training visualizations in section 4.5 of the revised manuscript, clearly showing that CLUSMol successfully emphasizes meaningful community structure while simultaneously maintaining fine-grained molecule-level distinctions.
>
> **W3: FGMG Construction and Molecular Similarity.**
>
> * We would like to clarify that the FGMG is not constructed using direct pairwise similarity calculations, which would be computationally prohibitive for  large-scale datasets. Instead,  it leverages functional group information to form communities without exhaustive similarity calculations. We acknowledge the importance of validating the rationality of the resulting clusters and have added visualization results in section 4.5 of the revised manuscript. These results empirically demonstrate that the constructed graph and clustering effectively group semantically similar molecules in the representation space.
> ### References
> [1] You, Yuning, et al. "Graph contrastive learning with augmentations." Advances in neural information processing systems 33 (2020): 5812-5823.
>
> [2] Caron, Mathilde, et al. "Unsupervised learning of visual features by contrasting cluster assignments." Advances in neural information processing systems 33 (2020): 9912-9924.
>
> [3] Khosla, Prannay, et al. "Supervised contrastive learning." Advances in neural information processing systems 33 (2020): 18661-18673.

---

### Meta-Review · Area_Chair_znBE · 2026-01-03

**Summary:**

This paper proposes CLUSMOL, a cluster-assisted contrastive learning framework for molecular representation learning. The method introduces three main contributions: (1) Functional Group-Guided Molecular Graph (FGMG), which constructs molecular relationship graphs with linear complexity O(MHK) based on functional group substitutions; (2) Louvain clustering to generate pseudo-labels from the constructed graph; and (3) Chemical Semantic-Weighted Contrastive Loss (SemCoL) that treats molecules within the same cluster as "soft positive" pairs.

**Reviewer Concerns:**

Reviewer UWDD (4):
- Addressed concern: missing computational overhead analysis
- Outstanding concern: missing baselines, limited novelty, and scalability

Reviewer 5haP (4):
- Addressed concerns: clustering validation
- Outstanding concerns: limited novelty, justification for Louvain clustering

Reviewer PFAh (2):
- Addressed concerns: missing references, fair justification
- Outstanding concerns: activity cliffs, performance vs. SOTA

**Reviewer Scores:**

The reviewers would not have changed the score, since their concerns were not entirely addressed.

---

### Decision · Program_Chairs · 2026-01-26

Reject